# Sentinel Lymph Node Assessment in Endometrial Cancer: A Review

**DOI:** 10.3390/cancers16183202

**Published:** 2024-09-20

**Authors:** Christopher Clark, Vera Loizzi, Gennaro Cormio, Salvatore Lopez

**Affiliations:** 1Azienda Ospedaliera Universitaria “Policlinico di Bari”-Clinica di Ginecologia e Ostetricia, 70124 Bari, Italy; v.loizzi@oncologico.bari.it (V.L.); g.cormio@oncologico.bari.it (G.C.); 2Gynecologic Oncology Unit, IRCCS Istituto Tumori “Giovanni Paolo II”, 70124 Bari, Italy; salvatore.lopez@hotmail.com

**Keywords:** endometrial cancer, sentinel lymph node, lymphadenectomy

## Abstract

**Simple Summary:**

Sentinel lymph node assessment is becoming a standard of care procedure in patients with surgically treatable endometrial cancer due to its cost-effectiveness and the advantages it offers in guiding post-operative management. Unlike in breast cancer, however, several key aspects regarding this technique’s employment in endometrial cancer remain unclear, such as tracer injection volume and final pathology interpretation. The aim of this paper is to investigate the current literature on this technique in order to provide simple and clear insight on the matter and to facilitate the reproducibility of this technique, ultimately resulting in improving patients’ oncological outcomes.

**Abstract:**

As the number of patients diagnosed with endometrial cancer rises, so does the number of patients who undergo surgical treatment, consisting of radical hysterectomy, bilateral salpingo-oophorectomy, and bilateral pelvic lymphadenectomy or lymph node sampling. The latter entail intra- and post-surgical complications, such as lymphedema and increased intra-operative bleeding, which often outweigh their benefits. Sentinel Lymph Node (SLN) sampling is now common practice in surgical management of breast cancer, as it provides important information about the disease without jeopardizing surgical radicality and patient outcomes. While this technique has also been shown to be feasible in patients with endometrial cancer, there is little consensus on several aspects, such as tracer injection volume and site, pathological ultrastaging, and result interpretation. The aim of this review is to analyze the current literature on SLN assessment in order to help standardize the procedure.

## 1. Introduction

Endometrial cancer is one of the most common types of cancer worldwide, affecting primarily post-menopausal women in high-income western countries. Key risk factors are obesity, long-term exposure to high estrogen levels and sedentary behavior. It is estimated that about 68,000 new cases will be diagnosed in the US in 2024 alone. Most cases of endometrial cancer are diagnosed at an early stage and show a 5-year overall survival rate of over 80% [1].

The main prognostic factor in clinically early stages of endometrial cancer is lymph node involvement, thus making lymph node assessment a key point in surgical management of early-stage endometrial cancer [2]. Comprehensive lymphadenectomy not only increases operative time and blood loss, but it is also associated with surgical complications, such as blood vessel and nerve damage, lymphoedema, and lymphocyst formation.

The concept of the sentinel lymph node (SLN) was first introduced in 1960 [3]. The distinctive benefit of SLN mapping is the opportunity to avoid “over-staging”, leading to a relatively lower morbidity than full lymphadenectomy and the potential for improved diagnostic accuracy using ultrastaging [4].

Similarly to surgical management of early-stage breast cancer, SLN assessment has become a widespread technique and a valid alternative to systematic lymphadenectomy in early-stage endometrial cancer, as shown by its safety and accuracy, especially when combined with pathological ultrastaging.

There are, however, several pitfalls in SLN assessment in endometrial cancer, most notably the removal of presumed sentinel lymph nodes that on final pathology appear as adipose tissue or lymphatic trunks, and the fact that this technique’s cost may exceed its benefits in certain patient subgroups (i.e., patients with early-stage, non-invasive G1-G2 endometrioid endometrial cancer, in which risk of nodal spread is extremely low). Therefore, a standardized approach to the employment of SLN technique is needed. The aim of this review is to analyze the scientific knowledge about the current status of SLN assessment in endometrial cancer, evaluating different sentinel lymph node mapping techniques and their indications in endometrial cancer.

## 2. Materials and Methods

A comprehensive literature search on the retrieved publications (with earliest publication year set to 2008) was performed independently by four authors associated with this current study. The language of studies was limited to English only.

The literature search was conducted using the following electronic databases: Pubmed, Embase, Medline, and the Cochrane Library. The predefined keywords used for the search were “sentinel lymph node” and “endometrial cancer”. A search algorithm that selected and screened results based on a combination of the following search terms: “sentinel AND (endometri* OR uterus OR uterine OR corpus uteri) AND (cancer OR neoplasm* OR carcinoma* OR malignanc* OR tumo*)” was used to perform the literature search detailed in this study. We subsequently performed an ulterior screening of the retrieved publications in order to include only relevant results. In this review we included prospective observational cohort studies and retrospective studies only. Exclusion criteria for this study were: case reports, case series, and studies with low patient volume (<100 patients). A summary of the study method is provided in Figure 1.

## 3. Results

### 3.1. SLN Mapping and Failure Predictors

Different methods of SLN mapping have been used over the years since the introduction of the SLN technique. SLN mapping is performed using different kinds of substances, named “tracers”, which can be divided into dye-based and radioisotopic tracers. The former includes both fluorescent dyes, such as Indocyanine Green (ICG), and non-fluorescent dyes (i.e., blue dyes and Carbon NanoParticles—CNP). The latter are mainly comprised of metastable Technetium-99 (^99m^Tc).

ICG is a particular substance which produces fluorescent light when excited with infrared radiation and offers high sensitivity and specificity when used to detect lymph nodes. Blue dyes possess the property of permeating small caliber vessels shortly after injection and therefore provide rapid visualization of nodal stations. The main drawback in dye-based tracers, particularly in blue dyes, is the risk of allergic reaction, which can range from mild to anaphylaxis and must be considered in clinical practice.

^99m^Tc is a radioactive isotope which is often utilized in gynecological oncologic surgery for SLN mapping, particularly in apparently early-stage vulvar cancer in which the employment of this tracer is recommended by European guidelines for inguinal sentinel lymph node assessment [5]. Patient administration is carried out by nuclear medicine specialists, generally up to 24 h before surgery, and lymph node detection is made possible by pre-operative nuclear imaging or intra-operative use of gamma probes. Although this technique possesses high sensitivity and specificity, it exposes patients and healthcare providers to radiation and increases hospital costs both directly and indirectly by prolonging hospital admission.

A summary of different tracing techniques used in endometrial cancer, including overall detection rate, sensitivity, negative predictive value (NPV), and disadvantages is shown in Table 1.

The most-used substances in SLN mapping for endometrial cancer are Indocyanine Green (ICG) and blue dyes, as well as a combination of radiotracers and blue dyes. ICG has been shown to be superior to Isosulfan blue dye in detecting at least one sentinel node and is the standard dye in endometrial cancer SLN detection [10]. As regards tracer injection sites, different methods have been proposed in the literature. The most common sites of injection are the cervix and uterine fundus through hysteroscopy or laparoscopic injection [11,12,13]. The rationale for fluorescent dye injection at these sites lies in the uterus’ complex lymphatic drainage; in short, while the corpus uteri and isthmus have the same lymphatic drainage as the cervix (through the parametria to the iliac and obturator nodes located at the pelvic sidewalls), the fundus uteri’s lymphatic drainage directly involves the para-aortic nodes along the gonadal vessels [4,14], implying that SLN mapping through cervical injection of fluorescent dye may fail in endometrial cancer originating from the fundus uteri. However, it has been demonstrated that para-aortic metastases seldom occur in endometrial cancer, and mainly in high-grade endometrial cancer, non-endometrioid histologies and in stages >IB [15]. Moreover, Kumar et al. [16] conducted a retrospective study on 742 patients, 514 (70%) of which were considered “at risk” of lymph node metastasis (LNM). The authors’ analysis concluded that, in the absence of pelvic LNM, only 3% of patients show para-aortic LNM. This evidence suggests that hysteroscopic or laparoscopic injection of fluorescent dye in the fundus uteri may be omitted or at least used only in selected cases, as these techniques require long learning curves and increase procedure costs.

There is no consensus on the amount of fluorescent dye used to detect sentinel lymph nodes and whether ICG injection volumes influence SLN detection rates and SLN mapping failure. Taşkın S et al. [17] published a prospective observational cohort study in which 101 patients who underwent laparoscopic surgery for early-stage endometrial cancer were analyzed and divided into “Bilateral mapping” and “failed bilateral SLN mapping (unilateral or bilateral failed mapping)” groups, which were subsequently compared for demographic, clinical, surgical, and pathological features. The overall, unilateral, and bilateral SLN detection rates were 94.1%, 19.8%, and 74.3% respectively. The failed (unilateral or no mapping) bilateral detection rate was 25.7%. At multivariate analysis, failure rates were found to be higher in patients with greater cervical or uterine lengths, deep myometrial invasion and larger tumor size, but without statistical significance. Interestingly, BMI, and surgery type did not affect SLN mapping failure and increasing the ICG injection volume (4 mL vs. 2 mL) did not improve mapping rate significantly. A retrospective analysis carried out by Tortorella et al. [18] investigated predictors of unsuccessful SLN mapping using ICG in patients with apparent early-stage endometrial cancer undergoing surgical staging; 327 patients treated between 2014 and 2016 and on which SLN biopsy was attempted were retrospectively identified. SLN biopsy was successful in 256 of these patients (78.3%), while 71 (21.7%) had an unsuccessful procedure. At multivariate analysis, lysis of adhesions at the beginning of the procedure (OR 3.07, 95% CI, 1.56–6.07), as well as enlarged lymph nodes (OR 4.69, 95% CI, 1.82–12.11) were independently associated with SLN mapping failure. It is worth mentioning that in both cited studies, injection of <3 mL ICG also appeared to be predictive of failed SLN detection, although both authors agreed that this finding lacks statistical significance.

Another retrospective study [19] clarified the frequency of SLN locations using different volumes of ICG as SLN tracer. In this study, the authors retrospectively collected data from 352 patients who underwent radical hysterectomy for endometrial cancer between 2019 and 2023. Patients were divided into two groups depending on the volume of ICG used for SLN mapping: 2 mL group (1 mL injection superficially at hours 3 and 9 of the cervix) and 4 mL group (2 mL injection per side, 1 mL in the submucosal layer and 1 mL deeply). In the 2 mL group, the most common SLN locations were: external iliac (75%), obturator fossa (14%), common iliac (10%), and sacral (1%); in the 4 mL group, the Authors described SLN detection in the external iliac region (66%), obturator fossa (22%), common iliac (7%), sacral (3%), parametrial (1%), and aortic (1%). This study also showed that a volume of 1 mL ICG injected superficially at h 3 and 9 of the cervix can be used for detection of SLN in early-stage endometrial cancer, whereas a volume of 2 mL per side should be considered in obese patients or in other settings, such as FIGO 2009 stages > IB or in cases of gross cervical involvement, as this volume allows for better bilateral detection. It must be noted that higher ICG volumes can lead to excessive fluorescent signals in the parametrium or retroperitoneum: as demonstrated by the authors, the detection of fluorescent signal outside of the external iliac nodes group (especially in the obturator fossa, sacral, parametrial, and aortic lymph node groups) was higher in patients who received a 4 mL ICG cervical injection.

Another paper investigating factors related to SLN mapping failure was published by Body et al. [20]. In this study, a total of 119 patients who underwent surgery for endometrial cancer from 2014 to 2015 were included. Patient demographics, surgical technique, and histopathological findings were collected prospectively. Univariate analysis was performed to evaluate factors associated with SLN mapping failure: overall and bilateral detection rates were 93% and 74%, respectively. Advanced disease stage (FIGO III-IV) was the only factor related to SLN mapping failure (*p* = 0.01).

Sozzi et al. [21] investigated the topic of failed SLN mapping by retrospectively analyzing data from patients who underwent laparoscopic surgery for apparent early-stage endometrial cancer from 2016 to 2019 in four different institutions. Exclusion criteria included evidence of lymph node involvement in preoperative workup, synchronous invasive cancer, the use of tracers different from ICG, and neoadjuvant therapy. In total, 376 patients were included in the final analysis, with an overall, bilateral, and unilateral SLN detection rate of 96.3%, 76.3%, and 20%, respectively. The failed bilateral mapping detection rate was 23.7%. At multivariate analysis, Lymph-Vascular Space Invasion (LVSI) [OR 2.4 (1.04–1.12), *p* = 0.003], non-endometrioid histology [OR 3.0 (1.43–6.29), *p* = 0.004], and intra-operative finding of enlarged lymph nodes [OR 2.3, (1.01–5.31) *p* = 0.045] were predictive of failed SLN mapping.

Based on the evidence obtained by these studies, enlarged lymph nodes, lysis of adhesions at the beginning of surgery, FIGO stage III-IV, non-endometrioid histology, and LVSI status remain the most important predictors of SLN mapping failure. It is unclear whether ICG injection volume influences detection rates; it should be noted that high ICG injection volumes could lead to excessive fluorescent signal and subsequent failure in SLN detection, although higher volumes could be considered in obese patients, FIGO 2009 stages > IB and in cases of gross cervical involvement; on the other hand, ICG injection volumes < 3 mL (although not statistically demonstrated) could lead to mapping failure.

### 3.2. SLN Ultrastaging Techniques

SLN ultrastaging is a technique which combines Hematoxylin and Eosin (H&E) staining and immunohistochemistry (IHC) on final pathology in order to detect low-volume metastatic tissue in case of negative initial histological evaluation. Although SLN ultrastaging is an increasingly used technique to assess SLN status in endometrial cancer, there is no consensus on optimal assessment method: notably, in breast cancer SLN, disease classification is based on the largest cluster of contiguous tumor cells, not considering any distance between clusters [22]. In endometrial cancer literature, this is not explicit, thus leading to variable interpretation of metastatic disease.

Sarah Chiang et al. [23] have recently reported their experience with ultrastaging at Memorial Sloan Kettering Cancer Center. In the authors’ experience, SLN are examined by performing sections at 2 mm intervals parallel to the longest axis and stained with hematoxylin and eosin. If no sentinel lymph node is found, no isolated tumor cells are detected or if micro- or macrometastasis are detected, no additional workup is required; if a positive SLN is detected, ultrastaging is carried out in high risk histotypes (serous, clear cell, FIGO grade 3 endometrioid, and undifferentiated/dedifferentiated carcinoma and carcinosarcoma), myo-invasive tumors, or, in case of isolated tumor cells, detection.

When required, ultrastaging is performed by cutting two adjacent 5 μm paraffin block sections at each of two levels, 50 μm apart (named L1 and L2). These are then stained with H&E and cytokeratin AE1:AE3 immunohistochemistry. SLN diagnosis is defined by visual estimation of disease extent measuring the largest cluster of contiguous tumor cells in a single cross section. Unfortunately, there are no standardized protocols for SLN assessment in endometrial cancer, as demonstrated by the wide range of SLN detected by ultrastaging, which clearly affect adjuvant therapy recommendations. With these premises, the authors proposed a standardized SLN assessment protocol for endometrial cancer. In order to do so, records of 285 patients with newly diagnosed endometrial cancer and positive SLN treated only with primary surgery from January 2013 to January 2020 were collected. SLN slides subjected to ultrastaging were also obtained and digitalized for retrospective analysis. Depending on the cluster’s size or the number of tumor cells, SLN metastasis was defined as macrometastasis (>2 mm), micrometastasis (>0.2 mm but ≤2 mm or >200 cells) or isolated tumor cells (ITC) (≤0.2 mm or <200 cells).

After excluding patients with positive SLN and no ultrastaging, as well as excluding patients with incomplete digitalization of L1 and L2 slides, 109 patients were included in the final dataset. Concordant SLN diagnoses on L1 and L2 slides were seen in 91.8% of SLNs, suggesting that one immunohistochemical slide was diagnostic in most patients, although cost-effectiveness was not directly investigated in this study. Digital slide review down-graded metastatic carcinoma to isolated tumor cells in 4.1% of SLNs, reflecting inter-observer variability and improved diagnostic accuracy using digital pathology measuring tools.

### 3.3. Management of Positive SLN, ITC, Micrometastases, and Macrometastases

About 40% of patients with positive SLN experience recurrence of disease after standard treatment [24]. However, many factors influence the risk of recurrence in patients with positive SLN, such as LVSI status and lymph node metastasis size. Moreover, ultrastaging techniques have shown that more than 50% of patients with positive SLN have so-called “low-volume metastases”, for which there is no consensus regarding clinical implication.

ESGO/ESTRO/ESP guidelines recommend adjuvant chemotherapy and/or external beam radiation therapy (EBRT) with or without vaginal brachytherapy in patients with local and/or regional cancer spread (stage IIIA–IIIB) and pelvic/paraaortic lymph node metastasis (stage IIIC) [25]. Hence, it appears clear that patients with lymph node macrometastasis benefit from adjuvant treatment. According to the new FIGO 2023 staging of endometrial cancer, SLN micrometastasis detected with ultrastaging is considered as metastatic involvement (pN1(mi)) [26]. Despite this, there currently is no consensus on whether patients with micrometastases could benefit from adjuvant strategies. A recently published retrospective multicenter registry-based study by Ignatov et al. [27] investigated the relationship between nodal micrometastases and clinical outcomes in 428 patients, who were sorted into three groups: 302 (70.6%) patients were node negative and did not receive adjuvant treatment; 95 (22.2%) patients had nodal micrometastases and received adjuvant treatment; and the last 31 (7.2%) patients were diagnosed with nodal micrometastases, but did not receive adjuvant treatment after surgery. The authors found that patients with nodal micrometastases who received adjuvant therapy had comparable Disease-Free Survival (DFS) to that of patients with negative-node metastases (*p* = 0.648), and most importantly patients with micrometastatic nodes who did not undergo adjuvant therapy had significantly worse DFS compared to the aforementioned groups (*p* = 0.0001). Therefore, there is strong evidence suggesting that adjuvant therapy significantly reduces recurrence risk in patients with micrometastases and thus should be offered to this subgroup of patients.

The role of ITC in orienting post-surgical therapies is more controversial. An interesting multicenter retrospective study published in 2021 by Backes et al. [28] included 175 patients with ITC diagnosed on SLN and otherwise stage I-II with endometrioid histology. Associations between treatment modalities, tumor characteristics, and Recurrence-Free Survival (RFS) were evaluated. Of these patients, 76 (43%) did not receive adjuvant therapy or received vault brachytherapy alone. A total of 21 (12%) received EBRT, and 78 (45%) received chemotherapy with or without radiation. It must be noted that most of the patients who received systemic treatment had high-risk tumor features, such as LVSI+, myoinvasion and high-grade disease. Median follow-up time was 31 months. In total, 9 of the 175 patients (5.1%) had cancer recurrence after a variable time lapse. At Cox proportional hazard models, adjuvant treatment was not associated with RFS (HR = 0.63, 95% CI 0.11–3.52, and HR = 0.90, 95% CI 0.22–3.61, respectively). These findings suggest that cancer recurrence in patients with ITC metastases is rare, regardless of treatment strategies. Moreover, positivity for ITC alone was not the only criterion used to define further treatment strategies. This should always be the case, as ITC alone does not represent valid prognostic factors in predicting cancer recurrence, and treatment plans should be tailored to the tumor’s biological features (Lymph Vascular Space status, myoinvasion, grading, etc.).

### 3.4. Sentinel Lymph Node Evaluation in Atypical Endometrial Hyperplasia

Atypical endometrial hyperplasia is a well-known precursor of invasive endometrial carcinoma. Over a 5-year time period, 25% of patients with a diagnosis of atypical endometrial hyperplasia progress to endometrial cancer, although the existing literature suggests that about 30–50% patients who undergo surgical treatment for atypical hyperplasia have concurrent endometrial cancer in final pathology specimens [29,30]. This can be explained by multiple factors, such as low specimen volume obtained during endometrial biopsies, the absence of endometrial stroma in biopsy specimens, and the chance of not targeting the cancerous lesion during hysteroscopic biopsy procedures. This raises the question as to whether patients treated with upfront surgery for atypical endometrial hyperplasia may benefit from lymph node status assessment. Rosati et al. [31] published a multicentric retrospective study analyzing the role of SLN assessment in patients with atypical hyperplasia, with the scope of improving prognostic and therapeutic information in this large group of patients. The authors also compared surgical adverse events in patients undergoing simple hysterectomy vs. hysterectomy plus SLN biopsy in order to assess whether the risk of SLN biopsy exceeds the theoretical benefits of this procedure in these patients. The results of this study show a comparable estimated blood loss, as well as intra- and post-operative complications among the two groups, while Sentinel Lymph Node Biopsy added relevant prognostic and therapeutic information in 60.8% of patients. While the authors also acknowledge that the vast majority of patients (71.4%) with concurrent endometrial carcinoma and atypical hyperplasia have low-risk endometrial cancer (low-grade, non-myometrial-invading, and endometrioid histology) on final pathology specimens, thus making the role of SLN evaluation in these patients debatable, it is also important to note that SLN biopsy added important information for modulation of adjuvant therapy in 12.3% of patients with high to intermediate risk of concurrent endometrial carcinoma, without increasing surgery time and/or complications. This study also prompts the need for further investigation in order to predict the presence of invasive endometrial carcinoma in patients with atypical endometrial hyperplasia during pre-operative diagnostic work-up.

### 3.5. Sentinel Lymph Node in Apparently Early-Stage Endometrial Cancer

As already pointed out, it is debatable whether every patient with endometrial cancer may benefit from SLN assessment, as certain subgroups have low risk of nodal involvement. This is especially true for low-grade, non-aggressive subtypes of endometrial carcinoma [32,33]. Given the low risk of Lymph Node Metastasis (LNM) in this subset of patient, Sentinel Lymph Node biopsy could prove a safer staging method as compared to Pelvic/paraaortic Lymph Node Dissection (PLND/PALND), reducing risk of intra- and post-operative complications.

The SENTI-ENDO study [9] was designed to assess SLN biopsy detection rate and accuracy in endometrial cancer. In this prospective multicenter cohort study, 133 patients with FIGO stage I-II endometrial cancer were enrolled at nine centers around France from 2007 to 2009. All patients were administered cervical injection of technetium colloid and patent blue prior to primary surgery. SLN biopsy (SLNB) was carried out for each patient, followed by systematic pelvic node dissection. At least one SLN was found on final histology in 111 patients; 19 of 111 (17%) had pelvic lymph node metastasis, while 5 of 111 (5%) had associated para-aortic metastasis. When the hemipelvis was considered as unit of analysis, Negative Predictive Value (NPV) was 100% (95% CI 95–100) and sensitivity 100% (95% CI 63–100), whereas considering the patient as unit of analysis, only three patients (all of whom had type 2 endometrial cancer) had false negative results (NPV 97%, 95% CI 91–99; sensitivity 84%, 95% CI 62–95). With these findings, the authors concluded that SLNB could be considered as a valid alternative to PLND in patients with early stage, low-risk endometrial carcinoma. Moreover, 10% of low-risk and 15% of high-risk endometrial cancer patients were upstaged through SLNB, suggesting that this technique could prove useful in tailoring adjuvant therapy.

Another more recent study [6] examined the role of SLNB technique in surgical staging of patients with early-stage endometrial cancer. A total of 385 patients with clinical stage I endometrial cancer were enrolled between 2012 and 2015 in different institutions across the USA. Some 340 patients underwent SLN mapping and PLND, while PALND was performed in 196 of these patients. Ultimately, 293 patients (86%) had successful mapping on at least one sentinel lymph node. A total of 41 patients (12%) had positive SLN, and 36 of these had at least one mapped SLN. Finally, 35 of these 36 patients had nodal metastasis identified in the sentinel lymph nodes (sensitivity 97.2%, 95% CI 85–100; negative predictive value 99.6% 95% CI 97.9–100), thus confirming previously observed results.

A multicenter retrospective study conducted by De Vitis et al. [34] examined the risk of nodal involvement in patients who underwent primary surgery for apparent early-stage endometrial cancer. In this study, the authors evaluated the risk of nodal involvement and stratified patients based on tumor features, considering the implications of the new FIGO 2023 classification of endometrial carcinoma. SLN pathological evaluation and ultrastaging techniques were applied as discussed above. This study found that, in accordance with the new FIGO 2023 classification of endometrial cancer, some histological subtypes of endometrial carcinoma have higher risk of nodal involvement even in apparent early stages: therefore, SLN assessment is of utmost importance in these cases, as it improves staging accuracy and offers better treatment strategies. This is especially true in the case of high-grade endometrioid endometrial cancer and serous endometrial cancer, whereas low-grade endometrioid endometrial cancer has low risk of nodal involvement and, as suggested by ESGO-ESTRO-ESP 2020 guidelines, not only SLN assessment, but even lymph node sampling can be omitted in these cases.

SLN ultrastaging remains controversial in low-grade endometrioid carcinoma with <50% myometrial invasion, especially in G1 tumors, although authors suggest applying this technique unless the impact of isolated tumor cells is considered to be irrelevant, due to the fact that, in these patient subgroups, a risk of nodal involvement of about 3% (1.5% for grade 1 endometrial cancer) is estimated.

While all these studies suggest the feasibility of SLN biopsy as an alternative to PLND/PALND in patients with early-stage endometrial cancer, data regarding oncologic outcome in terms of disease-free survival, overall survival and patient outcomes are scarce. One retrospective study [35] evaluated two cohorts of patients with BMI ≥ 35 kg/m^2^ (patients who underwent LND vs. patients who underwent SLN without pelvic systematic lymphadenectomy) treated between 2007 and 2017 for endometrial cancer. The median operative time for patients in the SLN group was shorter than that of patients in the LND group and the SLN group had lower median estimated blood loss than the LND group. After 24 months, 98% of the patients were alive and 95.5% were disease-free, without differences in terms of Overall Survival (OS), Progression-Free Survival (PFS), and Disease-Specific Survival (DSS). Another retrospective study [36] reported no difference in Recurrence-Free Survival (RFS) in patients who underwent SLN mapping and ultrastaging vs. patients treated according to French guidelines for presumed low- or intermediate-risk endometrial cancer. Although encouraging, these results are based on retrospective observation, while prospective data are scarce; currently, at least two studies [37,38] are evaluating oncological outcomes of patients with apparently early-stage endometrial cancer treated with radical hysterectomy and SLN mapping.

### 3.6. Sentinel Lymph Node Biopsy in High-Grade Endometrial Carcinoma

The vast majority of studies regarding SLN evaluation in patients with endometrial cancer include patients with low-grade endometrioid endometrial carcinoma, which are known for lower risk of lymph node metastasis, whereas the role of SLN biopsy in patients with high-grade, aggressive subtypes of endometrial carcinoma is less known. This is exemplified by examining the main international guidelines regarding sentinel lymph node mapping: Dick et al. [39] conducted a descriptive comparative study of the National Comprehensive Cancer Network (NCCN), the Society of Gynecologic Oncology (SGO), the European Society of Gynecological Oncology (ESGO), the British Gynecological Cancer Society (BGCS), and the Japan Society of Gynecologic Oncology (JSGO) guidelines regarding the topic of SLN in endometrial cancer. While most societies agree on the use of ICG dye for SLN mapping, ICG injection site, performing SLN ultrastaging, performing side-specific lymphadenectomy when SLN mapping fails, and considering SLN mapping as an alternative to full lymphadenectomy in low-intermediate risk stage I-II endometrial cancer, indications in the case of high-risk endometrial cancer differ between organizations. Notably, SGO guidelines consider SLN mapping in high-risk endometrial cancer feasible with completion of full lymphadenectomy (LAD) and para-aortic assessment, whereas ESGO states that SLN mapping is an acceptable alternative to full LAD in stages I-II [25,40,41]. Therefore, according to the main gynecological oncology societies’ recommendations, SLN mapping is less advocated for high-risk endometrial cancer, and is at most “accepted” as an alternative to LAD.

A large multicenter prospective cohort study by Rossi et al. [6] examined patients with clinical stage I endometrial cancer of all histologies and grades undergoing robotic surgery and receiving a standardized cervical injection of ICG for SLN mapping followed by PLND and PALND in a timespan from 2012 to 2015. Three hundred and eighty-five (385) patients were enrolled in this trial. PLND was performed in all patients, while PALND was performed in 74 out of 100 patients with high-grade tumors. 293 patients had a successful SLN mapping. The sensitivity of the sentinel lymph node technique to identify nodal metastatic disease was 97.2% (95% CI 85.0–100: McNemar’s *p* = 1). Among the 258 patients with negative sentinel lymph node results, 257 had truly negative non-sentinel lymph nodes, resulting in a negative predictive value of 99.6% (95% CI 97.9–100). This study shows that SLN mapping using ICG has high degrees of diagnostic accuracy in detecting endometrial cancer metastases, and that SLN biopsy can safely replace PLND in endometrial cancer staging. One important flaw of this study is that, although 28% of the FIRES study population had high-grade histologies, which are at highest risk for metastases and isolated para-aortic metastases, the role of Sentinel Lymph Node Biopsy in these highest-risk patients is not definitively addressed in this study population. It should also be noted that only one false negative result occurred in the study, and this occurred in a patient with high-grade endometrial cancer. The authors also state that while SLN biopsy can be considered as a valid alternative to PLND and PALND in endometrial cancer, this technique should be applied with algorithms that account for failed mapping cases.

Cusimano et al. [42] examined the accuracy of SLN biopsy as compared to lymphadenectomy in patients with intermediate-to-high-risk endometrial cancer. This prospective study is one of the largest studies in which patients with low-risk endometrial cancer were excluded from selection. In total, 156 patients who underwent surgery for stage I G2 endometrioid or high-grade endometrial cancer (EC) from 2015 to 2019 were included in the final dataset. All patients underwent SLN biopsy followed by lymphadenectomy: more specifically, patients with G2 endometrioid EC underwent pelvic lymphadenectomy alone, whereas patients with high-grade EC underwent PLND and PALND. A total of 30 (19.2%) had grade 2 endometrioid EC, and 126 (81%) had high-grade EC. Sentinel lymph node detection rates were 97.4% per patient (95% CI, 93.6–99.3%), 87.5% per hemipelvis (95% CI, 83.3–91.0%), and 77.6% bilaterally (95% CI, 70.2–83.8%). Twenty−seven patients (17%) had metastatic disease in their sentinel lymph node or lymphadenectomy specimens, twenty-four had high-grade EC, and only three had grade 2 endometrioid EC.

In total, 26 of 27 patients with node-positive disease were correctly identified by the Sentinel Lymph Node Biopsy (SLNB) algorithm, yielding a sensitivity of 96.3% and a Negative Predictive Value (NPV) of 99.2%, suggesting that SLN mapping is comparable to PLND and PALND in lymph node metastasis detection even in high-grade endometrial cancer. Furthermore, even the authors of this study acknowledge that if the SLN technique is to be used in this subset of patients, a strict algorithm incorporating both side-specific PLND and PALND in case of SLN mapping failure, which commonly occurs in patients with high-grade endometrial cancer due to lymphatic drainage damage, must be considered.

### 3.7. Further Perspectives and Ongoing Studies

Currently ongoing research aims to establish new SLN detection methods by prospectively recording data of patients who undergo minimally invasive surgery. A phase-II, open-label, randomized pilot study aims to compare SLN detection rate in two groups of patients randomized to receive Transvaginal Ultrasound-guided Myometrial Injection (TUMIR) of [^99m^Tc] Tc-albumin nanocolloid radiotracer vs a combination of radiotracer + ICG the day before surgery. A planar and Single Photon Emission Computed Tomography (SPECT-TC) scan will be carried out in order to obtain the patients’ lymphatic maps, in order to evaluate and compare sentinel lymph node number and drainage territories. On the day of surgery, all patients will receive intracervical injection of methylene blue at the start of surgery and SLN detection will take place both via Near InfraRed (NIR) laparoscopic optics and gamma probes. After one month, the proportion of patients with intraoperative detection of sentinel nodes will be measured, as well as other secondary endpoints such as: (i) the number of sentinel nodes biopsied during surgery after injection of the [^99m^Tc] hybrid radiotracer Tc-albumin nanocolloid-ICG (hybrid RT) or albumin nanocolloid (RT) radiotracer [^99m^Tc] by TUMIR, (ii) the number of sentinel lymph nodes with infiltration detected during surgery, and (iii) the difference in the number of sentinel nodes and drainage patterns visualized after injection of the hybrid RT or RT between TUMIR lymphogammagraphy and cervical lymphogammagraphy. Another ongoing prospective, single-arm interventional study aims to assess the feasibility of a double-tracer injection technique (cervical injection of ICG + subserosal injection of charcoal carbon black dye) in patients with early-stage endometrial cancer in order to reduce para-aortic sentinel lymph node failure rates. An interesting ongoing clinical trial aims to evaluate the role of real-time visualization of lymphatic flow in patients who undergo robotic surgery for endometrial cancer. Patients are administered intracervical injection of ICG while the surgeon observes real-time ICG appearance in lymphatic vessels transperitoneally using the robot’s firefly mode: the aim of this study is to compare data from patients who undergo this procedure with data from another group of patients in which standard SLN assessment is carried out, so as to determine any difference in SLN detection rates (Table 2).

Two ongoing prospective studies aim to assess the usefulness and stability of SLN mapping in early-stage (SELYE) and high-risk (ALICE) endometrial cancer, respectively. The first study’s primary endpoint is to measure three-year progression-free survival in two groups of patients with early-stage endometrial cancer who underwent laparoscopic or robotic surgery and either SLN biopsy or routine lymph node dissection. The second study’s design is very similar, as it aims to obtain data on oncological outcomes in patients with high-risk endometrial cancer who undergo SLN biopsy vs routine lymphadenectomy (Table 3).

The SENECA study (Staging Endometrial caNcer Based on molEcular ClAssification) is a retrospective study which aims to assess the role of molecular classification in predicting lymph node status in patients with early-stage endometrial cancer who underwent surgery. The results of this large study could help to understand the clinical implications of the new FIGO 2023 molecular classification of endometrial cancer and possibly provide useful information in tailoring SLN assessment to different molecular types of endometrial cancer. Another prospective study with results expected in 2026 aims to validate the use of a molecular panel of estrogen-induced genes to predict recurrence in low-risk endometrial cancer. To this aim, 500 patients with early-stage, low-risk endometrial cancer will be enrolled and subjected to standard surgical treatment. Before and after surgery, patients will also undergo collection of blood samples for tumor marker analysis. At the time of surgery, the collected specimens will be sent for molecular testing. Among other secondary endpoints of this study, the predictive ability of molecular panels in lymph node involvement will be assessed. Hopefully, this research will shed light on establishing different patterns of nodal recurrence based on different molecular subtypes of endometrial cancer, allowing for different mapping techniques to be used and reducing failure rates in the assessment of SLN (Table 4).

Another interesting subject worth investigating is the use of new tracers in SLN assessment. In fact, as stated above, many tracers are available in clinical practice for SLN mapping and provide high detection rates, sensitivity, and specificity. There are, however, several aspects which can limit the accuracy and safety of the aforementioned tracers; for example, the accuracy of ICG is inversely proportional to surgery time and requires intensive training in order to be carried out safely and effectively.

A major breakthrough in SLN mapping is represented by magnetic tracers. Superparamagnetic iron oxide (SPIO) consists of magnetic particles which can be systemically administered and subsequently phagocytosed by macrophages and transported into lymph nodes, where they can be detected by Magnetic Resonance Imaging (MRI). MRI detection of lymph node metastases using SPIO has been reported in patients with various oncological conditions, including breast cancer [43]; however, although this technique has been shown to be effective in breast cancer, there is no evidence on the effectiveness of magnetic tracers in uterine cancer.

A recent review summarized the most recent findings regarding the use of SPIO as an SLN tracer in breast cancer [44]. The study concluded that magnetic tracers are not inferior to combined radionuclide and dye-based tracing in node detection for breast cancer. Moreover, according to the study’s findings, SPIO tracing appears generally safe to use, with main health concerns being skin staining resulting from the procedure. Theoretical advantages of using SPIO instead of classic tracers are avoiding dye-tracers-related anaphylaxis and radiation damage by radionuclides, although there are no studies which compare side-effects between SPIO and other tracers. Furthermore, the cost-effectiveness of this method as compared to other mapping techniques was not assessed. A pilot study by Murakami et al. [45] evaluated the use of SPIO as a tracer for SLN detection in patients with uterine cancer through the injection of this substance in the cervix of 15 patients with uterine (cervical or endometrial) cancer scheduled for lymph node dissection. The study concluded that SPIO and radioisotopes are uptaken by sentinel lymph nodes in similar proportions in patients with uterine cancer. Although these findings are encouraging, the small number of cases assessed in this paper does not allow us to draw definitive conclusions; in our opinion, further research is needed in order to safely include SPIO in clinical protocols for SLN mapping in endometrial cancer.

## 4. Discussion

Sentinel Lymph Node assessment is a fairly new practice in endometrial cancer management; thus, its widespread employment is subject to each center’s experience. The aim of this review is to provide the most up-to-date information regarding SLN mapping techniques and indications and ideally standardize the procedure in order to provide patients with the best possible care.

ICG has been shown to be the best fluorescent dye for SLN mapping, given its sensitivity, low costs, and low rate of adverse events compared to other tracers (i.e., methylene blue or radionuclides), and should be the main choice when available. As regards injection sites, an h3 and h9 injection in the superficial layer of the cervix offers the best outcomes in terms of lateral wall lymph node mapping. The learning curve associated with cervical injection of tracer is also much less steep if compared to other injection sites (fundus uteri via hysteroscopy or laparoscopy) and provides better information, as lymphatic drainage to the lateral wall lymph nodes is much more common than para-aortic drainage. This technique also reduces operative time and thus may result in better intra- and post-operative outcomes.

The amount of fluorescent dye to inject in order to obtain acceptable SLN mapping is not clear. One study showed that different volumes of ICG can alter SLN mapping sites: higher volumes of ICG are associated with SLN detection in the parametrial and para-aortic lymph node groups. Higher rates of SLN mapping failure occur using <3 mL of ICG, although this finding lacks statistical power. In our opinion, prospective studies comparing tracer injection volumes and SLN failure mapping rates are needed in order to standardize this aspect. Another important topic we believe deserves more in-depth analysis is the adaptation of ICG volumes to patients’ BMIs in order to reduce failure rates in this broad subset of patients. In our institution, we utilize a standard dose of 2 mL ICG injected at h3 and h9 of the cervix prior to abdomen incision, regardless of the patient’s BMI, in accordance with Memorial Sloan Kettering SLN mapping protocol. In our experience, increasing tracer volume in obese patients does not increase the rate of SLN detection, while often resulting in failed mapping due to the tracer spreading to the parametria. These findings, however, may be due to the surgeon’s own experience and need statistical confirmation.

The most important predictors of SLN mapping failure are advanced-stage disease (FIGO III-IV), LVSI, non-endometrioid histology, pre- or intra-operative finding of enlarged lymph nodes, and lysis of adhesions during surgery. Given that most of these findings cannot be excluded in routine pre-operative workup, a standard protocol of systemic pelvic lymph node dissection should be kept in mind and applied in case of SLN mapping failure (both unilateral and bilateral) in high-risk patients in which lymphatic drainage could be altered. Another important aspect we would like to stress is that SLN mapping failure rate increases with longer operation time; thus, it is important that surgeons performing SLN mapping for endometrial cancer receive intense training before attempting surgery on high-risk patients (i.e., obese patients who have history of abdominal surgery, PID, peritonitis, or any other conditions which may result in abdominal adhesion formation).

Ultrastaging techniques should be employed in the case of negative findings on standard H&E staining in final pathology, and ideally standardized in order to reduce bias and improve reproducibility. While post-surgical management of patients with macrometastasis is quite straightforward (adjuvant chemotherapy and/or external beam radiation therapy (EBRT) +/− vaginal brachytherapy in patients with stage IIIA to IIIC in accordance with NCCN and ESGO/ESTRO/ESP guidelines), it must be noted that adjuvant therapies significantly increase disease burden and should be carefully considered in patients with micrometastasis or ITC; while there are studies suggesting that adjuvant therapy may have a role in patients with micrometastasis, the employment of chemoradiation in patients with ITC is more controversial. According to the findings of several studies, ITC alone should not guide decisions for adjuvant therapies.

An interesting aspect regarding SLN metastasis derives from the new FIGO 2023 classification of endometrial cancer: although several authors auspicated that the new molecular classification could harbor new understandings of endometrial cancer biology (particularly on low-volume nodal metastasis prediction and management) [46], data on this aspect are scant. Original research by Schivardi et al. [47] retrospectively identified 317 patients with endometrial cancer who underwent radical surgery and subsequent molecular characterization from April 2019 to December 2021 in the European Institute of Oncology, Milan. These were divided into four groups according to the tumors’ molecular profiles: 150 (47.3%) had Non-Specific Molecular Subtype (NSMP), 101 (31.9%) were MisMatch Repair deficient (MMRd), 38 (12%) had p53 abnormality (p53abn), and 28 (8.8%) had mutation in POLE exomerase domain (POLEmut). Among them, 64 (20.2%) had lymph node metastasis, including 29(45.3%) NSMP, 26(40.6%) MMRd, 8(12.5%) p53abn, and 1 (1.6%) POLEmut. At univariate analysis, POLEmut vs. other risk classes was found to have a protective role (*p*: 0.03) in lymph node metastasis, thus suggesting that surgical evaluation of pelvic lymph nodes through SLN mapping could be safely avoided in patients with the POLEmut molecular profile. However, limited experience in attempting to correlate low-volume disease with molecular and genomic profile failed to demonstrate an impact on outcomes of patients with low-volume disease. Furthermore, given the retrospective nature of this paper, conclusions were drawn on data relative to patients who already underwent surgical treatment and whose molecular profiles were obtained on final pathology; in our opinion, a priori knowledge of POLE status would be required in order to guide surgical management in this subset of patients. The current level of evidence is too scarce to draw any conclusions regarding the role of nodal disease in patients with POLE mutation, and further studies are needed to demonstrate any correlation between conventional risk factors and molecular/genomic characterization of endometrial tumors. The role of SLN mapping is debatable in patients with endometrial atypical hyperplasia, a well-known precursor of endometrial carcinoma. Although most patients with atypical endometrial hyperplasia progress to low-grade endometrial carcinoma, it is important to note that some patients with atypical endometrial hyperplasia may have concurrent intermediate-to-high-risk endometrial cancer; SLN mapping in these patients provides valuable information in defining subsequent treatment and thus should be considered. These findings, however, reflect the current challenges in detecting invasive cancer in patients with precursor lesions and thus prompt the need for tailoring present diagnostic tools in order to exclude the presence of invasive cancer in pre-operative diagnostic workup.

An interesting contribution to the field of SLN mapping in endometrial carcinoma is derived from radiomics, whose role in managing patients with endometrial cancer has been recently proposed. Liu et al. [48] investigated the role of radiomics in predicting Lymph Node Metastasis (LNM) status in early-stage endometrial carcinoma patients in order to aid in clinical decision making. The authors conducted a retrospective study in which 707 patients with clinically early-stage endometrial cancer were randomly divided into “training” and “test” cohorts. All patients underwent MRI and radiomics features were extracted from the imaging files. The researchers built three models: clinical (grade 1–2 endometrioid tumors by dilatation and curettage and less than 50% myometrial invasion on MRI without cervical infiltration), radiomics (selected radiomics features), and radiomics nomogram (combining selected radiomics features, myometrial invasion on MRI, and cancer antigen 125). Receiver Operating Characteristics (ROC) curves were used to assess the predictive performance of the three models, and clinical decision curves, net reclassification index (NRI), and total integrated discrimination index (IDI) were also calculated. Results showed that the predictive radiomics nomogram was able to predict LNM and therefore assist in surgical decision making in patients with clinically early-stage endometrial carcinoma, suggesting that such models could compete or be integrated with sentinel lymph node assessment in surgical management of endometrial cancer.

Further studies are also needed in order to obtain information regarding the oncological outcome of patients with endometrial carcinoma who underwent surgical mapping of sentinel lymph nodes; while current data on this topic is mainly derived from retrospective analyses, prospective knowledge in terms of PFS and OS in these patients is lacking. The results of at least two prospective studies on this matter are expected in the next few years and we believe these will be crucial in improving our understanding of this complex disease.

## 5. Conclusions

The burden of endometrial carcinoma on women’s health and healthcare systems around the world is increasing. While new research thrives and sheds new light on understanding this complex disease’s biology, several aspects of managing patients with endometrial cancer remain unclear. This paper summarizes current evidence on SLN mapping techniques and pathological ultrastaging of nodal disease in order to improve and standardize this procedure. To our knowledge, there are no studies in the literature which explore the correlation between SLN mapping and EC molecular classification. Our research highlights the importance of molecular profiling of endometrial cancer, as this could potentially influence surgical treatment. Moreover, we strongly believe that gynecological cancer treatment greatly benefits from multidisciplinary management; therefore, in our opinion, implementing ultrastaging protocols in this review could help other Institutions in standardizing care for endometrial cancer patients.

The main weakness of our study lies in the fact that most of the papers found in the literature surrounding endometrial cancer and sentinel lymph node sampling are retrospective; therefore, data regarding surgical and oncological outcomes of EC patients are scarce. This limits our attempt to provide clear information regarding SLN mapping and therefore establish easily applicable guidelines. Further research is needed to clarify the role of SLN biopsy in improving oncological outcomes and the correlation between endometrial cancer molecular classification and surgical management.

## Figures and Tables

**Figure 1 cancers-16-03202-f001:**
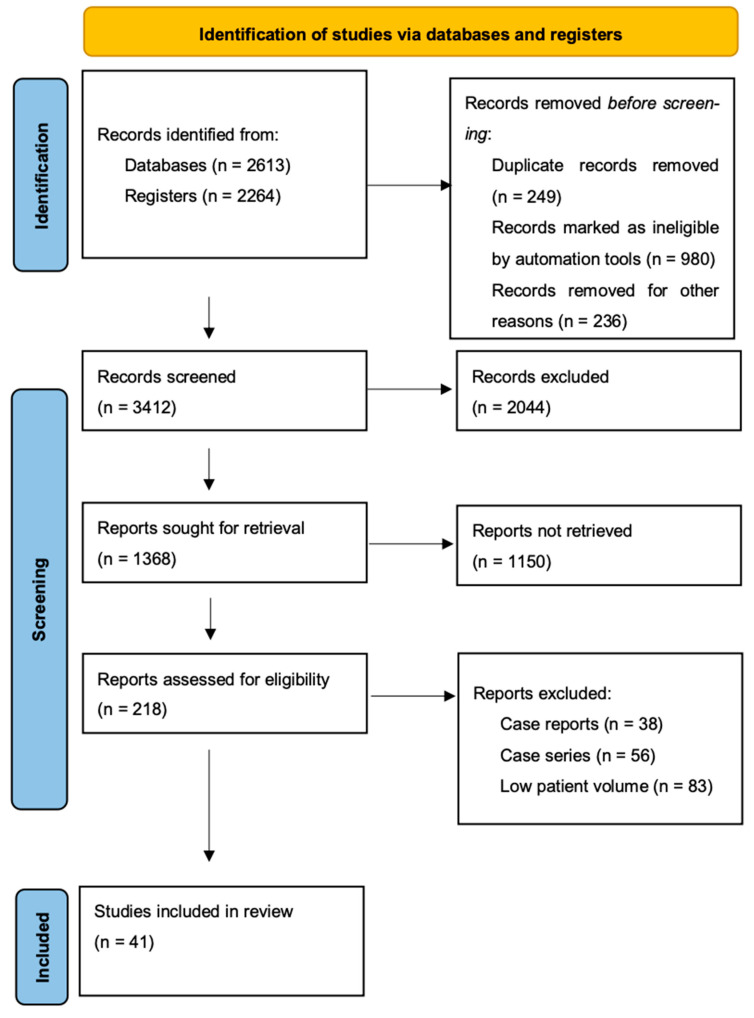
Flow diagram of study selection.

**Table 1 cancers-16-03202-t001:** comparison between ICG and combined radioactive and blue-dye tracers.

Tracer	Overall Detection Rate	Sensitivity	Negative Predictive Value	Disadvantages
ICG	93% ^†^	97.2% *	99.6% *	High costs, long learning curve, needs near-infrared light source for detection, mapping failure due to prolonged operating times and inexperienced surgeons
Blue Dyes (BD) + ^99m^Tc	89% **	84% ^§^	100% **	Patient discomfort, higher rates of adverse reactions, exposure to radiation, longer operating time

*: adapted from [6]. ^†^: adapted from [7]. ^§^: adapted from [8]. Available from: https://ijgc.bmj.com/content/24/6/1048.abstract (accessed on 1 August 2024). **: adapted from [9].

**Table 2 cancers-16-03202-t002:** Ongoing interventional studies focusing on new SLN mapping strategies.

NCT	Title	Study Type	Intervention	Status
NCT04492995	Sentinel Node in Endometrial Cancer (HYBRIDENDONOD)	Interventional	Diagnostic Test: injection via TUMIR of the tracer [^99m^Tc] Tc-albumin nanocoloid-ICG (6 mCi, 4 mL) Diagnostic Test: injection via TUMIR of [^99m^Tc] Tc-albumin nanocoloid (6 mCi, 8 mL).	Recruiting
NCT06163963	Sentinel Lymph Node in Early-Stage Endometrium Cancer	Interventional	Diagnostic Test: Sentinel Lymph Node Mapping With Double Tracer and Double Injection Sites in Early-Stage Endometrium Cancer	Recruiting
NCT05191212	The Role of Real-time Appearance of Lymphatic Flow in Lymphatic Mapping in Endometrial Cancer	Interventional	Procedure: indocyanine green	Recruiting

**Table 3 cancers-16-03202-t003:** Ongoing interventional studies assessing oncological outcomes of SLN mapping in early-stage and high-risk endometrial cancer.

NCT	Title	Study Type	Intervention	Status
NCT04845828	Randomized Comparison Between Sentinel Lymph Node Biopsy and Lymph Node Dissection in Early-Stage Endometrial Cancer (SELYE)	Interventional	Procedure: Sentinel lymph node mapping Procedure: Routine lymph node dissection	Recruiting
NCT03366051	Sentinel Node Mapping in High-Risk Endometrial Cancer (ALICE)	Interventional	Procedure: Sentinel Node Mapping Procedure: Lymphadenectomy	Recruiting

**Table 4 cancers-16-03202-t004:** Ongoing observational studies assessing the predictability of endometrial cancer metastatic sites based on the new FIGO 2023 molecular classification of endometrial cancer.

NCT	Title	Study Type	Groups and Intervention	Status
NCT05707312	Staging Endometrial caNcer Based on molEcular ClAssification	Observational	POLE mutant endometrial cancer patients Mismatch repair endometrial cancer patients No specific mutational profile endometrial cancer patients to NSMP P53 abnormal endometrial cancer patients	Recruiting
NCT04604613	Prediction of Recurrence Among Low-Risk Endometrial Cancer Patients	Observational	Bilateral Salpingectomy with Oophorectomy Biospecimen Collection Hysterectomy Other: Laboratory Biomarker Analysis Correlative studies Lymph Node Mapping Sentinel Lymph Node Biopsy	Recruiting

## Data Availability

The original data presented in the study are openly available in PubMed at https://pubmed.ncbi.nlm.nih.gov (accessed on 1 August 2024).

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
