# Peer review of "Sentinel Lymph Node Assessment in Endometrial Cancer: A Review"

_cancers, 2024, doi:10.3390/cancers16183202_

Round 1

Reviewer 1 Report

Comments and Suggestions for Authors

The study presented for review is an interesting review of available data on the role of sentinel lymph node identification and evaluation in endometrial cancer. The manuscript has an adequate structure and the discussion is very fluid. My concerns and suggestions for improvement for this manuscript are as follows:

1. please provide definition for all abreviation before first usage in text, and correct minor English mistakes. 

2. Please complete the methodology by naming inclusion and exclusion criteria for the studies included. An adequate search-flow diagram according to PRISMA should be included. 

3. Why limit the search to literature after 2008? The method is older than that.

4. A comparison between all SLN techniques should be included perhaps in tabelated form, comprizing of detection rate, false-positives, false-negatives and oncologic outcomes. Please include data about radiactive and magnetic methods of SLN mapping also.

5. A comparison between SLN mapping by cancer type would also be a great addition (endometrial vs other gynecologic, breast and other malignancies.

6. Please provide diferences between this study and any other already published. Also add strenghts and limitations.

Comments on the Quality of English Language

Minor editing needed.

Author Response

Specific comments:

  1. please provide definition for all abreviation before first usage in text, and correct minor English mistakes.

Response: following Reviewer’s suggestion, minor English mistakes have been identified and corrected. Full definition for all abbreviation have been added before first usage in text.

  1. Please complete the methodology by naming inclusion and exclusion criteria for the studies included. An adequate search-flow diagram according to PRISMA should be included.

Response: A more detailed description of study selection and exclusion criteria has now been added in the “materials and methods” section. Moreover, as suggested by the Reviewer, a search-flow diagram according to PRISMA has been included in the “materials and methods” section (referred to in text as: Figure 1).

  1. Why limit the search to literature after 2008? The method is older than that.

Response: the 2008 date filter was used in order to include the most up-to-date information regarding Sentinel Lymph Node biopsy in endometrial cancer. Moreover, the study from Benedetti et al, published in 2008, concluded that lymphadenectomy did not improve overall survival in patients with early stage endometrial carcinoma while increasing morbidity through formation of lymphocysts, lymphedema, and varying degrees of short and long-term neuralgia. This study has subsequently prompted research in the direction of sentinel lymph node assessment as opposed to lymphadenectomy in patients with clinically early stage endometrial cancer; we believe that selecting studies based on the same year in which this very important study was published strongly helped in screening results.

  1. A comparison between all SLN techniques should be included perhaps in tabelated form, comprizing of detection rate, false-positives, false-negatives and oncologic outcomes. Please include data about radiactive and magnetic methods of SLN mapping also.

Response: following Reviewer’s suggestion, we have now included a short paragraph addressing different sentinel lymph node mapping technique in the “results” section (lines 131-167). A short table reporting differences between ICG and combined 99mTc and blue-dye tracers has also been added (referred to in text as: Table 1).

Furthermore, an additional paragraph focusing on paramagnetic tracers (lines 672-707) has now been included in the “further perspectives and ongoing studies” section.

  1. A comparison between SLN mapping by cancer type would also be a great addition (endometrial vs other gynecologic, breast and other malignancies.

Response: as Reviewer suggested, we have included in-text references to different malignancies and SLN mapping strategies. However, rather than including an additional subchapter focusing on the differences between SLN mapping in endometrial carcinoma and other malignancies, we preferred limiting these references to brief descriptions of the different strategies so as to not drive the attention away from the main topic.

  1. Please provide diferences between this study and any other already published. Also add strenghts and limitations.

Response: a brief description of our study’s strength and limitations, as well as differences with other already published studies, has now been included in the “conclusions” subchapter (lines 839-850).

Reviewer 2 Report

Comments and Suggestions for Authors

As indicated by the author the manuscript is focused on sentinel lymph node assessment for helping standardization of the procedure in endometrial cancer. The authors suggested that sentinel lymph node assessment is important but there is little consensus on several aspects, such as tracer injection volume and site, pathological ultrastaging and result interpretation in endometrial cancer. Therefore, in order to provide simple and clear insight on the matter and to facilitate the reproducibility of this technique, the authors analyzed the current literature on sentinel lymph node assessment. Overall, the paper is clearly written and interesting to read. It specifically findings in the manuscript will be important for precision medicine of endometrial cancer patient. However, I need to mention that the study seems a lacking authors’ own opinion. Nevertheless, to my mind such a review is worth publishing.  

Author Response

comment 1: I need to mention that the study seems a lacking authors’ own opinion.

response: Following Reviewer’s suggestion, we have now implemented a small paragraph concerning our centre’s experience with sentinel lymph node assessment in the “discussion” section (lines 732-738).

Reviewer 3 Report

Comments and Suggestions for Authors

This work is nicely written, very comprehensive and put an up-to-date information available for the readers. However, it is mainly focused on ICG technique results and some other potential tracers could be discussed (only minor references to radio tracers or blue dyes). Although the current guidelines enhance the use of ICG, some limitations of this tracer (empty packets, high BMI patients, more sentinel nodes number depending on time, etc..has not been discussed.

The subchapters are very well chosen and the information is well characterized.

Some minor typos regarding sentinel nodes instead of SLN.

REFERENCES must be corrected and adapted to the journal´s rules, as there are a lot of different descriptions and names must be ordered accordingly to the guidelines (Vancouver or similar).

Author Response

comment 1: This work is nicely written, very comprehensive and put an up-to-date information available for the readers. However, it is mainly focused on ICG technique results and some other potential tracers could be discussed (only minor references to radio tracers or blue dyes). Although the current guidelines enhance the use of ICG, some limitations of this tracer (empty packets, high BMI patients, more sentinel nodes number depending on time, etc..has not been discussed.   response: following Reviewer’s suggestion, we have now implemented a brief description of other tracers in the “results” section. Furthermore, we implemented a short table (referred to in text as “Table 1”) in which we also summarize the limitations of the most used tracers   comment 2: REFERENCES must be corrected and adapted to the journal´s rules, as there are a lot of different descriptions and names must be ordered accordingly to the guidelines (Vancouver or similar).   response: we have now corrected and adapted references using Vancouver citation style.

Round 2

Reviewer 1 Report

Comments and Suggestions for Authors

The authors have addressed all my concerns adequately and I thank them for their efforts. The manuscript can be published in the current form.